# Integrated accounts of behavioral and neuroimaging data using flexible recurrent neural network models

**Amir Dezfouli**[12#] **Richard Morris**[3†] **Fabio Ramos**[3‡] **Peter Dayan**[4§] **Bernard W. Balleine**[1*]

[1]UNSW Sydney [2]Data61, CSIRO [3]University of Sydney [4]Gatsby Unit, UCL

[#]akdezfuli@gmail.com [†]richardumorris@gmail.com [§]p.dayan@ucl.ac.uk
[‡]fabio.ramos@sydney.edu.au [*]bernard.balleine@unsw.edu.au

## Abstract

Neuroscience studies of human decision-making abilities commonly involve subjects completing a decision-making task while BOLD signals are recorded using fMRI. Hypotheses are tested about which brain regions mediate the effect of past experience, such as rewards, on future actions. One standard approach to this is model-based fMRI data analysis, in which a model is fitted to the behavioral data, i.e., a subject's choices, and then the neural data are parsed to find brain regions whose BOLD signals are related to the model's internal signals. However, the internal mechanics of such purely behavioral models are not constrained by the neural data, and therefore might miss or mischaracterize aspects of the brain. To address this limitation, we introduce a new method using recurrent neural network models that are flexible enough to be jointly fitted to the behavioral and neural data. We trained a model so that its internal states were suitably related to neural activity during the task, while at the same time its output predicted the next action a subject would execute. We then used the fitted model to create a novel visualization of the relationship between the activity in brain regions at different times following a reward and the choices the subject subsequently made. Finally, we validated our method using a previously published dataset. We found that the model was able to recover the underlying neural substrates that were discovered by explicit model engineering in the previous work, and also derived new results regarding the temporal pattern of brain activity.

## 1 Introduction

Decision-making circuitry in the brain enables humans and animals to learn from the consequences of their past actions to adjust their future choices. The role of different brain regions in this circuitry has been the subject of extensive research in the past [Gold and Shadlen, 2007, Doya, 2008], with one of the main challenges being that decisions – and thus the neural activity that causes them – are not only affected by the immediate events in the task, but are also affected by a potentially long history of previous inputs, such as rewards, actions and environmental cues. As an example, assume that subjects make choices in a bandit task while their brain activity is recorded using fMRI, and we seek to determine which brain regions are involved in reward processing. Key signals, such as reward prediction errors, are not only determined by the current reward, but also a potentially extensive history of past inputs. Thus, it is inadequate merely to find brain regions showing marked BOLD changes just in response to reward.

An influential approach to address the above problem has been to use model-based analysis of fMRI data [e.g., O'Doherty et al., 2007, Cohen et al., 2017], which involves training a computational model using behavioral data and then searching the brain for regions whose BOLD activity is related to the internal signals and variables of the model. Examples include fitting a reinforcement-learning model

to the choices of subjects (with learning-rates etc. as the model parameters) and then finding the brain regions that are related to the estimated value of each action or other variables of interest [e.g., Daw et al., 2006]. One major challenge for this approach is that, even if the model produces actions similar to the subjects, the variables and summary statistics that the brain explicitly tracks might not transparently represent the ones the hypothetical model represents. In this case, either the relevant signals in the brain will be missed in the analysis, or the model will have to be altered manually in the hope that the new signals in the model resemble neural activity in the brain.

In contrast, here, we propose a new approach using a recurrent neural network as a type of model that is sufficiently flexible [Siegelmann and Sontag, 1995] to represent the potentially complex neural computations in the brain, while also closely matching subjects' choice behavior. In this way, the model learns to learn the task such that (a) its output matches subjects' choices; and (b) its internal mechanism tracks subjects' brain activity. A model trained using this approach ideally provides an end-to-end model of neural decision-making circuitry that does not benefit from manual engineering, but describes how past inputs are translated to future actions through a successive set of computations occurring in different brain regions.

Having introduced the architecture of this recurrent neural network meta-learner, we show how to interpret it by unrolling it over space and time to determine the role of each brain region at each time slice in the path from reward processing to action selection. We show that experimental results obtained using our method are consistent with the previous literature on the neural basis of decision-making and provide novel insights into the temporal dynamics of reward processing in the brain.

## 2   Related work

There are at least four types of previous approach. In type one, which includes model-based fMRI analysis and some work on complex non-linear recurrent dynamical systems [Sussillo et al., 2015], the models are trained on the behavioral data and are only then applied to the neural data. By contrast, we include neural data at the outset. In a second type recurrent neural networks are trained to perform a task [e.g., to maximize reward; Song et al., 2017], but without the attention that we give to both the psychological and neural data. A third type aims to uncover the dynamics of the interaction between different brain regions by approximating the underlying neural activity (see Breakspear [2017] for review). However, unlike our protocol, these models are not trained on behavioral data. A fourth type relies on two separate models for the behavioral and neural data but, unlike model-based fMRI analyses, the free parameters of the two models are jointly modeled and estimated, e.g., by assuming that they follow a joint distribution [Turner et al., 2013, Halpern et al., 2018]. Nevertheless, similar to model-based fMRI, this approach requires manual model engineering and is limited by how well the hypothesized behavioral model characterizes its underlying neural processes.

## 3   The model

### 3.1   Data

We consider a typical neuroscience study of decision-making processes in humans, in which the data include the actions of a set of subjects while they are making choices and receiving rewards ($\mathcal{D}_{\text{BEH}}$) in a decision-making task, while their brain activity in the form of fMRI images is recorded ($\mathcal{D}_{\text{fMRI}}$).

Behavioral data include the states of the environment (described by set $\mathcal{S}$), choices executed by the subjects in each state, and the rewards they receive. At each time $t \in T^i$ subject $i$ observes state $s_t^i \in \mathcal{S}$ as an input, calculates and then executes action $a_t^i$ (e.g., presses a button on a computer keyboard; $a_t^i \in \mathcal{A}$ and $\mathcal{A}$ is a set of actions) and receives a reward $r_t^i$ (e.g., a monetary reward; $r_t^i \in \Re$). The behavioral data can be described as,

$$\mathcal{D}_{\text{BEH}} = \{(s_{t^i}^i, a_{t^i}^i, r_{t^i}^i) \,|\, i = 1...N_{\text{SUBJ}}, t^i \in T^i\}. \tag{1}$$

The second component of the data is the recorded brain activity in the form of 3D images taken by the scanner during the task. Each image can be divided into a set of voxels ($N_{\text{VOX}}$ voxels; e.g., 3mm x 3mm x 3mm cubes), each of which has an intensity (a scalar number) which represents the neural activity of the corresponding brain region at the time of image acquisition by the scanner. Images

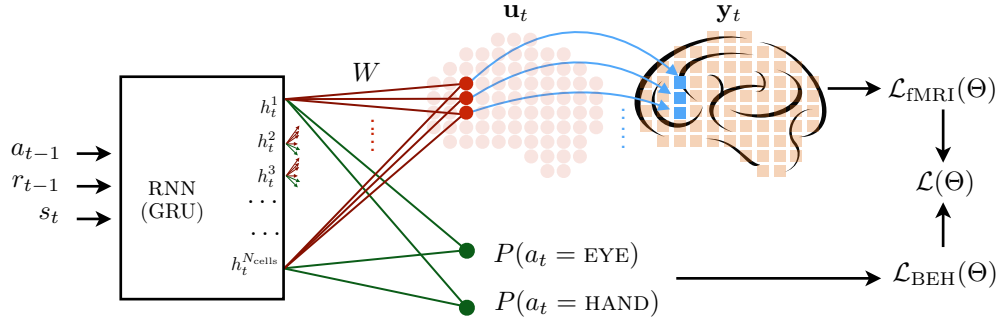

Figure 1: **Architecture of the model**. The model has a RNN layer which consists of a set of GRU cells, and receives previous actions, rewards and the current state of the environment as inputs ($N_{\text{CELLS}}$ is the number of cells in the RNN layer). The outputs/states of the RNN layer ($\mathbf{h}_t$) are connected to a middle layer (shown by red circles) with the same number of units as there are voxels ($N_{\text{VOX}}$); the outputs of the units $\mathbf{u}_t$ are weighted sums over their inputs. Each component of $\mathbf{u}_t$ is convolved with the HRF signal and is compared to the bias-adjusted intensity of its corresponding voxel in fMRI recordings ($\mathbf{y}_t$). Voxels are shown by the squares overlaying the brain, and three of them are highlighted (in blue) as an example of how they are connected to the units in the middle layer. The outputs of the GRU cells are also connected to a softmax layer (the green lines), which outputs the probability of selecting each action on the next trial (in this case, EYE and HAND are the available actions). $\mathcal{L}_{\text{BEH}}$ refers to the behavioral loss function and $\mathcal{L}_{\text{fMRI}}$ refers to the fMRI loss function. The final loss function is denoted by $\mathcal{L}(\Theta)$, which is a weighted sum of the fMRI and the behavioral loss functions. $\Theta$ contains all the parameters.

are acquired at times $0, \text{TR}, 2\text{TR}, \ldots, (N_{\text{ACQ}} - 1)\text{TR}$, where TR refers to the repetition time of the scanner (time between image acquisitions), and $N_{\text{ACQ}}$ is the total number of images. Let $y_t^{i,v}$ denote the intensity of voxel $v$ recorded at time $t$ for subject $i$. The fMRI data will take the following form:

$$\mathcal{D}_{\text{fMRI}} = \{y_t^{i,v}\}, t = 0, \text{TR}, 2\text{TR}, \ldots, (N_{\text{ACQ}} - 1)\text{TR}, i = 1 \ldots N_{\text{SUBJ}}, v = 1 \ldots N_{\text{VOX}}. \quad (2)$$

### 3.2 Network architecture

Actions taken by a subject at each point in time are affected by the history of previous rewards, actions and states experienced by the subject. Aspects of this history are encoded in neural activity in a persistent, albeit mutating, form, and enable subjects' future choices to benefit from past experience. This process constitutes learning in the task; we aim to recover it by jointly modeling the behavioral and neural data. We first describe the network architecture and then explain how it can be interpreted to answer the questions of interest.

**RNN layer.** The model (Figure 1) is a specific form of recurrent neural network (RNN). The recurrent layer consists of a set of $N_{\text{CELLS}}$ GRU cells [Gated recurrent unit; Cho et al., 2014]; cell $c$ outputs its state $h_t^c$ at time $t$. We define $\mathbf{h}_t$ as the state of the whole RNN network ($\mathbf{h}_t = [h_t^1, \ldots, h_t^{N_{\text{CELLS}}}]^\top$). This state summarizes the past history of the inputs to the network and is updated as new inputs are received according to a function that we denote by $f$,

$$\mathbf{h}_t = f(a_{t-1}, r_{t-1}, s_t, \mathbf{h}_{t-1}; \Theta), \quad (3)$$

depending on parameters $\Theta$. We aim to train the parameters of this dynamical system to approximate the underlying neural computations in the brain that translate previous inputs to future actions during the task.

**fMRI layer.** To establish a correspondence between the underlying RNN and neural activity, one training signal for $\Theta$ comes from requiring the activity of each voxel at each point in time to be described as a (noisy) linear combination of GRU cell states (shown by the red connections in Figure 1). We denote the weights of this linear combination as $W \in \Re^{N_{\text{VOX}} \times N_{\text{CELLS}}}$, and $\mathbf{u}_t$ as a vector of size $N_{\text{VOX}}$ representing predicted neural activity at each voxel at time $t$. Thus,

$$\mathbf{u}_t = W \mathbf{h}_t. \quad (4)$$

For training the model, the predicted neural activity is compared with the actual activity recorded by the scanner. However, neural activity is not instantly reflected in the intensity recorded by the scanner, but is delayed according to the haemodynamic response function (HRF; Figure S3). To correct for this delay, elements of $\mathbf{u}_t$ are first convolved with HRF [Henson and Friston, 2007], and after adding a bias term $\mathbf{b}$, are compared with the intensities of the corresponding voxels, to form the following loss function,

$$\mathcal{L}_{\text{fMRI}}(\Theta) = \sum_t \|\mathbf{u}_t \circledast \text{HRF} + \mathbf{b} - \mathbf{y}_t\|^2, t \in \{0, \text{TR}, \dots (N_{\text{ACQ}} - 1)\text{TR}\}, \tag{5}$$

in which $\Theta$ is the model parameters ($W$ and RNN parameters), $\mathbf{y}_t$ is a vector of size $N_{\text{VOX}}$ containing the recorded activity of each voxel at time $t$. Symbol $\circledast$ is the convolution operator. The above loss function can be thought of as the logarithm of a Gaussian likelihood function. Note that in this case the convolution operator acts on the *output* of the network, and so this is not a conventional convolutional neural network, in which convolutions act on the *input*.

**Behavioral layer.** To ensure that the RNN also captures the behavioral data, a second training signal for $\Theta$ comes from requiring it to produce actions similar to those of humans. This is achieved by connecting the output of the RNN network to a softmax layer in Figure 1 (shown by the green lines), in which the weights of the connections determine the influence of each cell on the probability of selecting actions. Denoting by $\pi_t(a)$ the predicted probability of taking action $a$ at time $t$, we define the behavioral loss function as:

$$\mathcal{L}_{\text{BEH}}(\Theta) = - \sum_{t \in T'} \log \pi_t(a_t), \tag{6}$$

in which $T'$ refers to the timesteps at which the subject was allowed to execute an action.

**Training.** We define the overall loss function as the weighted sum of the behavioral and fMRI loss functions,

$$\mathcal{L}(\Theta) = \sum_{i=1}^{N_{\text{SUBJ}}} \mathcal{L}_{\text{BEH}}(\Theta; \mathcal{D}_i) + \lambda \mathcal{L}_{\text{fMRI}}(\Theta; \mathcal{D}_i), \tag{7}$$

with parameter $\lambda$ determining the contribution of the fMRI loss function, and $\mathcal{D}_i$ denoting the data of subject $i$. Note the above loss function can be thought of as the logarithm of the multiplication of a Gaussian likelihood function (for the fMRI part) – with $\lambda$ being related to the level of noise/variance in the likelihood function – and a multinomial likelihood function (for the behavioral part).

### 3.3 Interpreting the model

We seek to understand how the inputs to the network (previous rewards, actions, states) affect future actions through the medium of the brain's neural activity. Although different methods have been suggested for investigating the way the inputs to a neural network determine its outputs, the most fundamental quantity is the gradient of the output with respect to the input, which represents how much the output changes by changing the input (as used, for instance, by Simonyan et al. [2013] in the context of an image classification task).

Inspired by this, we defined two differential quantities relating rewards, actions and brain activity to each other. There are at least two 'layers' to this: off- and on-policy. In the off-policy setting, which is conventionally studied in model-based imaging, there is a *fixed* sequence of inputs, whose effects on *future* predicted probabilities and neural activities we determine. In the on-policy setting, which is used in settings such as approximate Bayesian computation [Sunnåker et al., 2013], future *choices*, and thus future inputs are also affected by past inputs. For the present, we consider the simpler, off-policy setting. This allows us to look, for instance, at the brain regions involved in mediating the effect of the reward that subject $i$ actually received at, say, time $t_1$ on the predicted probability of the action that the subject actually executed at, say, time $t_2$. For convenience, we drop notation for the fixed inputs for the subject; and indeed for the subject number (since we fit a single model to the whole group).

The first measure represents the behavioral effects of reward on future actions, which can be calculated as the gradient of the predicted probabilities of actions at each time $t_2$ with respect to the input received at time $t_1$. For the case of binary choices, which are the focus of the current experiment, with

EYE and HAND as the two available actions in the task, we only needed to calculate the probability for one of the actions. Let $\pi_{t_2}$ denote the probability of taking action EYE at time $t_2$. The effect of reward at time $t_1$ on the action at time $t_2$ can be calculated as follows,

$$d_{t_1,t_2}^{\pi r} = \frac{\partial \pi_{t_2}}{\partial r_{t_1}}.$$

This is a straightforward application of backpropagation (calculated using automatic differentiation), noting again that we consider the inputs received by the network between $t_1$ and $t_2$ to be fixed. $d_{t_1,t_2}^{\pi r}$ can be thought as capturing how much the probability of taking action EYE at time $t_2$ increases as the results of increasing the magnitude of reward earned at time $t_1$.

The second measure relates behavioral and fMRI data by exploiting the informational association between the predicted neural activity $\mathbf{u}_t$ and the state of RNN, $\mathbf{h}_t$. First, note that, at each time $t$, $\mathbf{h}_t$ is a Markov state for the RNN, in that *given* $\mathbf{h}_t$, RNN outputs after time $t$ are independent of their past. Thus, we can decompose:

$$\frac{\partial \pi_{t_2}}{\partial r_{t_1}} = \sum_{k=1}^{N_{\text{CELLS}}} \frac{\partial \pi_{t_2}}{\partial h_t^k} \frac{\partial h_t^k}{\partial r_{t_1}}, \quad \text{for any} \quad t \in \{t_1 + 1 \ldots t_2\}, \tag{8}$$

as the effect changing $r_{t_1}$ has on the predicted RNN state $h_t^k$ at time $t$, times the effect that a change in $h_t^k$ has on the action probability $\pi_{t_2}$ at $t_2$. Now, consider the case that $W^\top W$ is non-singular (note that $N_{\text{VOX}} \gg N_{\text{CELLS}}$). This implies that there is a one-to-one mapping between the RNN state and predicted neural activity:

$$\mathbf{h}_t = (W^\top W)^{-1} W^\top \mathbf{u}_t . \tag{9}$$

Thus, we can rewrite equation 8 in terms of the effect changing $r_{t_1}$ has on the predicted neural activity $u_t^v$ in each voxel at time $t$ times what a change in $u_t^v$ implies about a change in $\pi_{t_2}$, operating implicitly via what the change in $u_t^v$ tells us about a change in $\mathbf{h}_t$. We can write this as,

$$\frac{\partial \pi_{t_2}}{\partial r_{t_1}} = \sum_{v=1}^{N_{\text{VOX}}} \frac{\partial \pi_{t_2}}{\partial u_t^v} \frac{\partial u_t^v}{\partial r_{t_1}}, t = t_1 + 1 \ldots t_2. \tag{10}$$

Note that this is a correlational relationship – the direction of causality is from $\mathbf{h}_t$ to $\mathbf{u}_t$. Nevertheless the individual terms in this sum:

$$d_{v,t_1,t,t_2}^{\pi u r} = \frac{\partial \pi_{t_2}}{\partial u_t^v} \frac{\partial u_t^v}{\partial r_{t_1}}, \text{for any} \quad t \in \{t_1 + 1 \ldots t_2\}, \tag{11}$$

combine the influence that voxel $u_t^v$ at time $t$ receives from the reward at time $t_1$ (which is $\partial u_t^v / \partial r_{t_1}$), with the covariation between the voxel activity and the action at time $t_2$ (which is $\partial \pi_{t_2} / \partial u_t^v$). This quantifies the intermediation of voxel $u_t^v$ between the reward at $t_1$ and the action at time $t_2$.

We make two remarks: (i) The joint fitting of the model to both the behavioral and fMRI data was important that, if there are behaviorally equivalent solutions, then the one that can fit the neural data should be chosen; and (ii) for equation 10 to hold it is necessary for the state of the network to be fully determined by the neural activity ($\mathbf{u}_t$). This can hold in the case of GRU cells (provided that the hidden units do not partition into separate behavioral and neural groups). In contrast, in LSTM cells [Long short-term memory; Hochreiter and Schmidhuber, 1997], the cell states and cell outputs are different and are both required to determine the outputs in the next time-step, and therefore in the case of LSTM cells equation 10 does not hold.

## 4   Results

In this section we aim to show how the above measures can be used to study the neural substrates of decision-making in the brain.

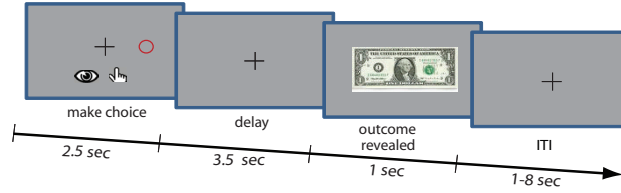

Figure 2: **The task**. Each trial started with the presentation of a screen (the left most square in the figure) and the subjects had 2.5 seconds to make an eye saccade to the red target circle or press a button with their right hand. After a delay (3.5 second) during which the screen showed only a fixation point, subjects received the outcome of their choice which indicated whether their choice was rewarded. The next trial started after an inter-trial interval (ITI) that varied between 1 and 8 seconds. Figure reprinted with permission from Wunderlich et al. [2009]. Copyright (2009) National Academy of Sciences.

## 4.1 Task and subjects

The data used here were previously published in Wunderlich et al. [2009]. The structure of the decision-making task is shown in Figure 2. In each trial subjects had a choice between making a saccade (EYE) or pressing a button (HAND). Choices were rewarded with varying probabilities across the experiment. There were four trial types in the task: (i) free-choice trials (150 trials), in which subjects could choose between EYE and HAND; (ii) forced-choice trials in which subjects were instructed to choose EYE (50 trials) or (iii) HAND (50 trials); (iv) null trials in which no reward was received irrespective of the action selected (50 trials). Forced-choices and null trails were randomly inserted between the free-choice trials. The environment consisted of two actions (EYE and HAND) and five states corresponding to the four trial types and one state when the choice outcomes were shown (reward or no-reward). Actions and states are assumed to be coded using one-hot representations. Since time was discretized (see below), there were time points at which no action was taken or no visual stimulus was shown, in which case states and actions were coded using zero vectors.

The total number of subjects was $N_{\text{SUBJ}} = 22$, and in total $N_{\text{ACQ}} = 1136$ images were acquired by the scanner each containing $N_{\text{VOX}} = 63191$ voxels. Therefore, the fMRI data can be summarized as a tensor of size $22 \times 1136 \times 63191$. Each subject made $\sim 300$ choices. See Supplementary Material for the details of fMRI preprocessing and model settings.

## 4.2 Model settings

All the methods were implemented in Tensorflow [Abadi et al., 2016] and gradients (for both optimization and interpretation of the model) were calculated using automatic differentiation methods available in this package. See Supplementary Material for the model settings.

## 4.3 From reward to action

Figure 3(a,b) shows two sets of off-policy simulations. In each simulation there are four choice states, the times of which are shown by the vertical gray patches in the top panels. The red patch following each grey ribbon shows the time at which the outcome was revealed following the choice. The first choice was rewarded (shown by 'R' in the graph), but the rest were not. In panel (a) action HAND was selected in all choice states whereas in panel (b) it was action EYE. Based on this, since in panel (a) the reward was earned when HAND was selected, we expected that choice to decrease the probability of selecting action EYE on the next choice. This is shown by the blue bars which illustrate the gradient of the probability of selecting the EYE action at each subsequent choice with respect to the amount of reward earned after the first choice ($d^{\pi r}$). For panel (b), since the reward was earned as a consequence of choosing EYE in the first choice, we expected the reward to have a positive effect on the probability of selecting EYE on the next trials, which is consistent with the graph.

Next we asked about the intermediation of each brain region between the reward earned after the first choice ($t_1$) and the next choice ($t_2$), shown by the red arrow in Figure 3(a). To answer this question,

we calculated $d^{\pi u r}$ for every voxel and every time-step between $t_1$ and $t_2$, and masked out the voxels that were not in the top one percent. By focusing only on the 99th percentile of $|d^{\pi u r}|$, we hoped to limit our analysis to the circuitry known to be involved in decision-making. The resulting voxel maps are shown in Figure 3(c) for the case of HAND action corresponding to the inputs shown in panel (a), and Figure 3(d) shows the time-course of changes in $d^{\pi u r}$. See Figure S1(c,d) for EYE action corresponding to the inputs shown in panel (b).

The results show that, for each action, the top 1% of voxels contain three key cortical and subcortical brain regions known to be critically involved in reward-processing and decision-making, i.e., (i) striatum (associative aStr; or ventral, vStr), (ii) anterior cingulate cortex (ACC) and (iii) supplementary motor area (SMA) [Rangel and Hare, 2010, Wunderlich et al., 2009]. We first note that these anatomical regions are among the same anatomical regions that Wunderlich et al. [2009] also identified as involved in decision-making in this task (see Figure S4 for the time course of changes in $d^{\pi u r}$ for the voxel coordinates reported in Wunderlich et al. [Table S3; 2009]).

Secondly, we can see that not only are the identified regions consistent with the neural substrates of decision-making based on previous work, but the temporal order of engagement of these regions is also consistent with their functional role in decision-making. It has been argued that activity in subregions of the striatum reflect reward prediction-errors [O'Doherty et al., 2004] and that these errors serve to update action-values in the ACC [Dayan and Balleine, 2002, Wunderlich et al., 2009, Seo and Lee, 2007, Walton et al., 2004], which in turn must be compared in the SMA to determine the best action before a decision can be made [Wunderlich et al., 2009]. Such prior work has argued that these different decision-making signals are carried by separate regions in a corticostriatal loop, which is assumed to participate in a time course of events leading to action-selection [Balleine and O'Doherty, 2010, Hare et al., 2011].

Here we show for the first time the temporal dynamics between these critical regions in the striatum, anterior cingulate cortex and motor areas leading to action-selection. Figure 3(d) shows the time course of each region's $d^{\pi u r}$ between the reward at 9.2 s ($t_1$) and the next response at 12.8 s ($t_2$). Note that since we took the probability of taking the EYE action as the reference, negative values of $d^{\pi u r}$ indicate a region's role in selecting the HAND action. At reward receipt (9.2 s), $d^{\pi u r}$ of the ventral striatum begins below the zero baseline and then (negatively) peaks at 9.8 s, as it mediates the effect of reward prediction-errors on the subsequent hand response. The value of $d^{\pi u r}$ for the anterior cingulate then (negatively) peaks after 10.4 s, consistent with its role in updating action values with the new errors before the next response. Finally $d^{\pi u r}$ for the large cluster in the motor area (including the supplementary motor area) controlling motor responses such as the HAND action, negatively peaks at the time of the action (12.8 s), which marks the end of the decision process in the current task.

As part of our supplementary material, Figure S1(d) shows the time dynamics between the striatum, anterior cingulate and motor areas controlling EYE choices – corresponding to the inputs shown in panel (b). Here positive values of $d^{\pi u r}$ indicate a region's role in selecting the EYE action. At reward receipt (9.2 s) the associative striatum is involved immediately in mediating the effect of reward on the subsequent action-selection. Then at 11 s the involvement of the anterior cingulate peaks before a region in the motor area nearest the supplementary eye field peaks at the time of action (12.8 s). In sum, changes in $d^{\pi u r}$ over this time period mirror those for the HAND action, and are consistent with the hypothesized roles of these regions in the varying decision stages of the reward-learning task used here.

## 5 Discussion

We have introduced a new neural architecture for investigating the neural substrates of decision-making in the brain. Unlike previous methods, our approach does not require manual engineering and is able to learn computational processes directly from the data. We further showed that the model can be interpreted to uncover the temporal engagement of different brain regions in choice and reward processing. Besides being used as a standalone analysis tool, this approach can inform model-based fMRI analyses to investigate whether the model correctly tracks the brain's internal mechanism. That is, if a brain region is found to be important in the current analysis, but not using the model-based fMRI analysis, this could mean that the model used to extract neural information is not representing all of the relevant neural signals involved in decision-making and requires further modification.

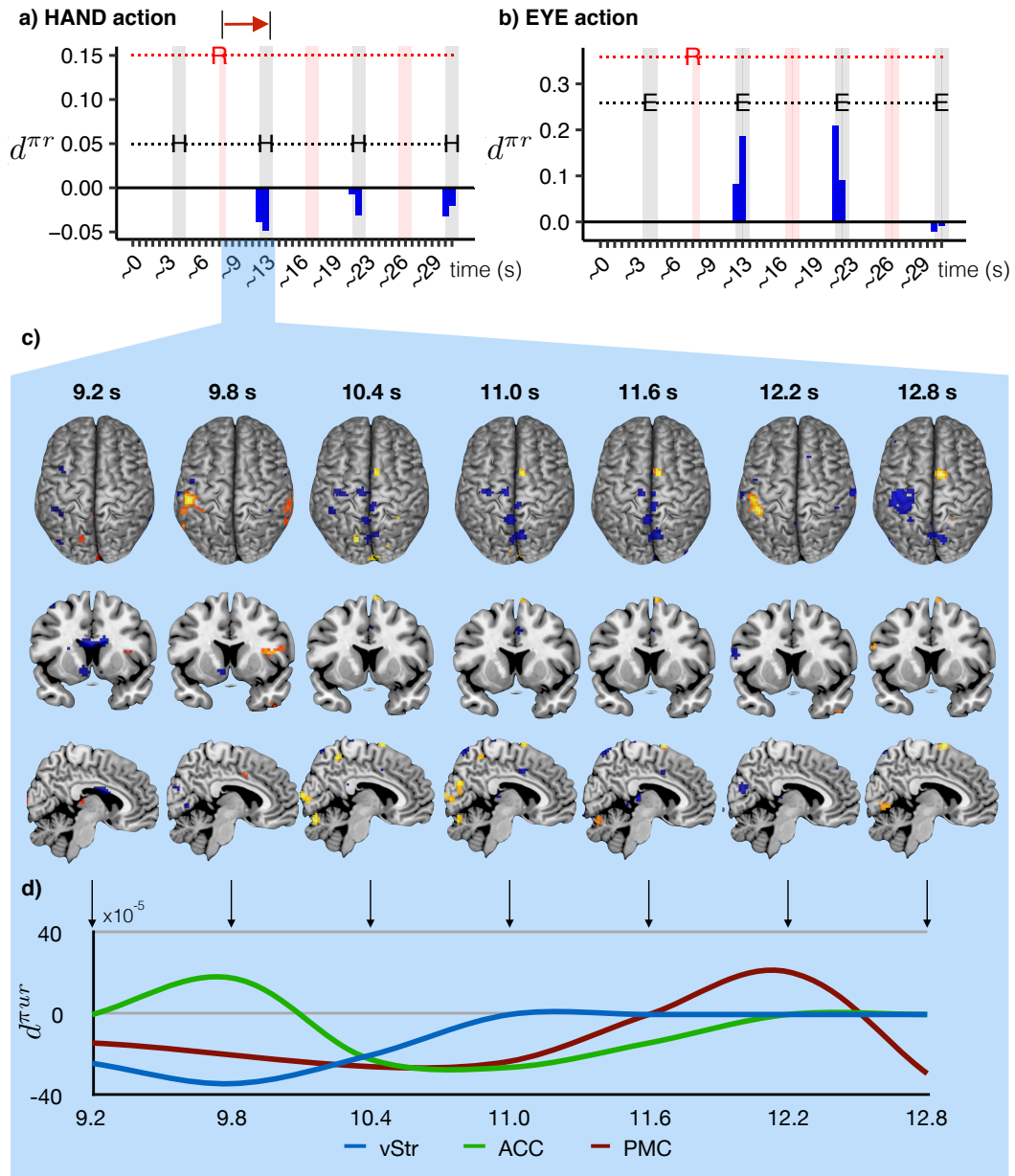

Figure 3: **(a,b)**. The graphs show the effect of reward on actions in terms of $d^{\pi r}$. The choice states (between EYE and HAND actions) are shown by the grey shaded area. In the left panel, action HAND (shown by 'H') was selected and in the right panel action EYE (shown by 'E') was selected at all of the choice states. The outcome of each choice (reward/no reward) was delivered in red shaded area. The first choice was rewarded, as shown by 'R' in the graph, but the other choices were not followed by any reward. The blue bars show the effect of reward received after the first choice on the subsequent choices ($d^{\pi r}$). **(c)**. Voxel maps and the time-course of changes in $d^{\pi ur}$ in cortical and subcortical brain regions between reward of the HAND action at 9.2 s and the response at 12.8 s shown by the red arrow in panel (a). Voxels below the 99th percentile of voxels were masked to reveal only the top one percent of voxels shown here. **(d)** The time courses of each region calculated from the maximum voxel in that region at each time point (smoothed), selected within an anatomical mask from wfu_pickatlas. $y$-axis represents $d^{\pi ur}$. ACC: anterior cingulate cortex; vStr: ventral striatum; PMC: primary motor cortex. See Figure S1 for the voxel maps and time course changes relating to the EYE action.

**Acknowledgments**

AD and BWB were supported by funding from UNSW Sydney and the National Health and Medical Research Council of Australia GNT1079561. PD was funded by the Gatsby Charitable Foundation. Part of this work was conducted whilst PD was at Uber Technologies. Neither body played a part in its design, execution or communication. PD is affiliated with Max Planck Institute for Biological Cybernetics, Tübingen, Germany (peter.dayan@tuebingen.mpg.de).

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
