[Supplementary Material · supplementary.pdf]

# Integrated accounts of behavioral and neuroimaging data using flexible recurrent neural network models

## Supplementary Material

Amir Dezfouli, Richard Morris, Fabio Ramos, Peter Dayan, Bernard W. Balleine

## S1 Model settings

All the methods were implemented in Tensorflow [Abadi et al., 2016] and gradients (for both optimization and interpretation of the model) were calculated using automatic differentiation methods available in this package. RMSProp [Tieleman and Hinton, 2012] and Adam [Kingma and Ba, 2014] optimization algorithms were investigated and RMSProp algorithm showed a faster convergence rate, and was used here.

The number of cells in the RNN layer was $N_{\text{CELLS}} = 48$, chosen based on computational constraints. The time of behavioral data was discretized with resolution $dt = 0.6625$. Similarly for the purpose of convolving the cells' outputs with HRF, the output times were discretized with a similar resolution $dt = 0.6625$. The choice of this specific $dt$ was because the TR of the scanner (time between consecutive image recordings) was $2.65$, which is divisible by $dt$, making it possible to perform the computations efficiently using the strided convolution operator.

For the purpose of regularization we took the following steps: (1) We first performed leave-one-out cross-validation at the subject level (leaving one subject out) using only the behavioral loss function to find the behavioral likelihood value on the training data that yielded the highest performance on the left-out subjects (Figure S2). (2) We used the likelihood value found in the previous step (adjusted for the number of subjects) to tune $\lambda$ using a search method. That is, we started with an initial value for $\lambda$ and performed the joint optimization (over the whole dataset); then we adjusted $\lambda$ based on the value to which the behavioral likelihood function converged in the previous iteration (i.e., decreased $\lambda$ if the behavioral likelihood converged to a value below the target value, and increased $\lambda$ otherwise), re-ran the joint optimization again using the new $\lambda$ value, and then iterated over this process. We were able to find the desired $\lambda$ with three iterations. This procedure encourages the network not to compromise on behavioral performance, while allowing it to choose amongst behaviorally equivalent solutions that fit the data best. Note that the same parameter setting was used for all the subjects. We ideally aimed to estimate $\lambda$ using a validation dataset, but because of the limited number of subjects here we used in-sample estimations for $\lambda$.

## S2 fMRI data

The details of fMRI data acquisition and preprocessing are described in Wunderlich et al. [2009]. In addition, the times-series of voxel intensities were passed through a high-pass filter with frequency 0.01Hz, and were also standardized to have a unit variance. Note that data for one of the subjects was not available and therefore 22 subjects were used in the current analysis instead of the 23 subjects used in Wunderlich et al. [2009].

HRF is approximated by the mixture of two Gamma functions with the parameters the same as the default parameters in 'spm_hrf' method in SPM package[1].

Figure S1: **(a,b)**. The graphs show the effect of reward on actions in terms of $d^{\pi r}$. The choice states (between EYE and HAND actions) are shown by the grey shaded area. In the left panel, action HAND (shown by 'H') was selected and in the right panel action EYE (shown by 'E') was selected at all of the choice states. The outcome of each choice (reward/no reward) was delivered in the red shaded area. The first choice was rewarded, as shown by 'R' in the graph, but the other choices were not followed by any reward. The blue bars show the effect of reward received after the first choice on the subsequent choices ($d^{\pi r}$). **(c)**. Voxel maps and the time-course of changes in $d^{\pi ur}$ in cortical and subcortical brain regions between reward of the EYE action at 9.2 s and the response at 12.8 s shown by the red arrow in panel (b). Voxels below the 99th percentile of voxels were masked to reveal only the top one percent of voxels shown here. **(d)** The time courses of each region calculated from the maximum voxel in that region at each time point (smoothed), selected within an anatomical mask from wfu_pickatlas. $y$-axis represents $d^{\pi ur}$. ACC: anterior cingulate cortex; aStr: associative striatum; SMA: supplementary motor area. See Figure 3 for the voxel maps and time course changes relating to the HAND action.

Figure S2: Cross-validation results for different numbers of optimization iterations using only the behavioral objective function ($\mathcal{L}_{\text{BEH}}$). Mean negative log-probability (NLP) averaged over cross-validation folds. Error-bars represent 1SEM.

Figure S3: **The haemodynamic response function (HRF).** The graph shows the relationship between the neuronal activities and recorded voxel intensities. Assuming that there was a neuronal activity at time $0$, the graph shows an approximation of how Blood Oxygenation Level Dependent signal (BOLD) changes over time. The BOLD signal corresponds to the voxel intensities $y_t^{i,v}$.

Figure S4: $d^{\pi\mu r}$ for the coordinates reported in Wunderlich et al. [Table S3; 2009]. For each coordinate and each time point, $d^{\pi\mu r}$ is calculated for all the voxels within 6mm of the coordinate and $z$-scores are calculated for each voxel (based on $d^{\pi\mu r}$ in the whole brain), and then $z$-scores are averaged among the voxels (within 6mm of the coordinate). The $z$-scores for the coordinates reported in Wunderlich et al. [Table S3; 2009] for (a) 'Vh' (b) 'Ve' (c) 'Vchosen' (d) 'Vchosen' (e) 'Vunchosen-Vchosen' (f) 'Vunchosen-Vchosen'. In panels (a), (d), (f), the simulation setting in Figure S1(b) is used (HAND action) and in the panels (b), (c), (e), the simulation setting in Figure S1(b) is used (EYE action).

## Footnotes

[1] https://github.com/spm/spm2/blob/master/spm_hrf.m