[Reviews · NeurIPS 2018]

Reviewer 1



In this paper the authors use a recurrent neural network model to study reward processing in human fMRI experiments. The main novelty of this approach is that it takes into account both the neural data (measured by fMRI) and the behavioral outcomes. In contrast the standard approach is to fit a computational model on the behavioral data, then use that as a regressor in a classical GLM analysis, which may find a good solution for the behavioral data but one that is different from the solution implemented in the brain. In this respect, this application stands out. Most other neural network models that are used to model fMRI data (e.g. Guclu and van Gerven, J Neuroscience 2015) do not use the fMRI data to train the neural network directly. The paper is well-written and clearly explained. The network topology the authors use is simple, elegant and appropriate for the proposed application. The implementation is fairly careful and the proposed method to 'unroll' the temporal dynamics to find the contribution of each timepoint and brain region to the reward signal (eq 11) is a nice way to visualise the underlying dynamics of the model, although not strictly novel because I am aware of other methods that use the derivatives of the predictive model for localising the discriminative signal. As such, I find this work to be highly original, very clearly written and of a sufficient methodological quality to be published at NIPS, contingent on a satisfactory response to the questions outlined below. I also think that the method could be adapted for other problems relatively easily so does make a significant contribution. My comments for improvement mainly relate to clarity: I would suggest to split the figures 3 and S1 and try to include all of them in the main manuscript (rather than splitting across main text and supplementary material). Switching from the different components in panels c and d of both respectively was initially quite confusing and I had to read the descriptions of both in the main text several times before I understood them. This could definitely be improved. The procedure for parameter optimisation part was not really clear (e.g. it is not really clear how Figure S2 is involved in parameter optimisation), nor is it clear which different settings for lambda (eq. 7) were evaluated. Was this done via a grid-search? How was leave-one-out cross-validation performed (e.g. subject level?). The authors state 'see the supplementary material for details of the optimization methods' but this only seems to refer to the optimizers chosen, not parameter optimization per se. Also it appears that a single cross-validation loop was employed, then the same parameter setting was used for all subjects. Is that correct? Please provide more details so that the reader can determine whether proper, unbiased parameter optimization techniques were employed.

Reviewer 2



This is an interesting paper advancing a new method of so-called “joint modeling”. The goal of this analysis is to create new neural models that are required to account for both brain data and behavior at the same time. This is a very important advance for cognitive neuroscience (indeed, it is almost necessarily the “right” way to approach neuroscience) and the approach developed here is novel to my understanding, interesting, well conducted, well described, and innovative. I think the paper could be improved in a couple of small ways. First is that joint modeling has actually be around for a while now and the authors review of past work was a little limited in that respect. I think the work by Tubridy, Gureckis and colleagues on joint models of memory/knowledge tracing are one example, but perhaps even more so B. Turner at Ohio State has been a pioneer in this idea. There certainly are sufficient differences between this past work and the present paper to mean that there is a lot of new ideas here and so I don’t mention these to suggest anything negative about the current paper. More that readers of this work might find it interesting to consider the broader space of ways people are thinking about this. I also had a little trouble understanding the interpretability of the model in terms of the gradients. I think a ML audience might find this more clear but for a general neuroscience/cog neuro audience what one is supposed to conclude from the gradients with respect to particular inputs is less transparent. Since this ends up being the main DV plotted in many of the figures I think it might help if the authors included a paragraph or so on what these things mean and the right want to think about them instead of just stating "the most fundamental quantity is the gradient of the output with respect to the input." In addition, I didn’t get a sense of the advantages of this model in terms of absolute prediction. I guess the main threshold I’d be interested in is if the model fit to both brain and behavior outperforms the model fit to behavior alone (or possibly more standard behavior only RL models). As best I can tell no model comparison is reported in the paper which is a major limitation on the contribution (and subsequent interpretability of the brain analyses/gradient terms). I totally believe the logic in the paper that the brain data might discriminate between equally compelling behavioral models but the results do not demonstrate this to be true here.

Reviewer 3



this paper proposes training a recurrent network to predict concurrent neuroimaging and behavior timeseries, using as input the stimuli and rewards seen by a subject in an experiment, along with the actions they emitted. although the model is trained on a combined supervised loss of predicting brain data and behavior, the weighting of these two factors is set such that the behavior takes precedence. because the state of the network is a restatement of the voxels, you can use calculus to find voxels 1) that are sensitive to reward (or the stimulus, although this isn't explored) and 2) to which the policy is sensitive. it seems like this method could be very powerful. however, because the model is so powerful, it's also kind of difficult to understand what's going on. suppose a reward happens at time t, and we consider an action at some later time t+T. in between, there is a series of voxel patterns. but is the RNN actually explaining anything about the computations happening in that series of voxel patterns? or is it just a fancy way of finding voxels that are related both to reward and action? i was also wondering about the lack of statistics to help understand how meaningful the patterns show in figure 3 are. the small size of the RNN is what prevents it from memorizing the data; it would be interesting to see how the results change as the capacity of the RNN changes. related to this i would be interested in more analysis of how the representational capacity of the model is divided between the brain and behavior. by the way lambda is set, it looks like the model fully explains the behavior (which is low dimensional). is it then the case that whatever parts of the brain data are correlated with behavior the model can then explain "for free"; and it uses its residual capacity to explain other parts of the brain data? why convolve the RNN's outputs with an HRF? shouldn't the network be able to learn the HRF, and maybe it would be more accurate (and region-specific) than a canonical HRF? since fMRI data has a lot of spatial structure, would it help to include a spatial deconv on the network's output?